# The Possibility of Using Bioelectrical Impedance Analysis in Pregnant and Postpartum Women

**DOI:** 10.3390/diagnostics11081370

**Published:** 2021-07-30

**Authors:** Aleksandra Obuchowska, Arkadiusz Standyło, Żaneta Kimber-Trojnar, Bożena Leszczyńska-Gorzelak

**Affiliations:** Department of Obstetrics and Perinatology, Medical University of Lublin, 20-090 Lublin, Poland; aobuchowska12@gmail.com (A.O.); zkimber@poczta.onet.pl (Ż.K.-T.); b.leszczynska@umlub.pl (B.L.-G.)

**Keywords:** bioelectric impedance analysis, body fat percentage, obesity, preeclampsia, cardiovascular disease risk, gestational weight gain, gestational diabetes mellitus

## Abstract

Pregnancy is a time of significant changes occurring in the composition of a woman’s body in order to provide support for the growth and development of the foetus. Bioelectrical impedance analysis (BIA) is used to assess the body composition and hydration status. This technique represents a non-invasive, reliable, and fast clinical approach, which is well tolerated by patients. A segmental impedance measurement might be advantageous in pregnant women, particularly in late pregnancy. The purpose of this paper is to provide a review of different applications of BIA in pregnant and postpartum women. It seems that BIA has a better prognostic potential for gestational and post-partum outcomes than body mass index. The BIA method can be successfully used to study the effect of excessive gestational weight gain in pregnancy on the development of obstetric complications. Studying the mother’s body composition and correlating it with her energy balance could facilitate the development of dietary recommendations for women. Evaluation of the body composition can provide important clues for diagnosis of gestational diabetes mellitus in pregnant women with a low risk of this disease. BIA is also used as one of the additional tests in assessing the risk of developing gestational hypertension and preeclampsia.

## 1. Introduction

Pregnancy is a time of significant changes occurring in the composition of a woman’s body in order to provide support for the growth and development of the foetus. Gestational weight gain (GWG) observed during pregnancy is associated with an increase in the maternal, foetal and placental tissue, changes in the amniotic and extracellular fluid, and blood volume expansion [1]. Despite the recommendations, only one in three pregnant women achieves healthy weight gain [2,3]. Appropriate GWG is important for the proper development of pregnancy as well as for avoiding complications such as, e.g., preeclampsia (PE) [1]. On the other hand, the puerperium is the period following delivery when the maternal physiological and anatomical changes return to the state from before the pregnancy. After delivery, breastfeeding lowers body mass index (BMI) and affects other adiposity measures. The clinical utility of body composition measurement in pregnancy is an ongoing area of research.

Bioelectrical impedance analysis (BIA) is used to assess the body composition and hydration status. This technique represents a non-invasive, reliable, and fast clinical approach, which is well tolerated and widely accepted by patients [4,5,6,7,8,9,10,11,12]. In the study of Bai et al., an excellent test repeatability throughout pregnancy has been demonstrated [13]. In spite of this, different bioimpedance measuring devices should not be used interchangeably [13]. Moreover, BIA measurements should not be taken when participants are dehydrated, within 4 h of consuming food and drink, have an unused bladder, and within 12 h of intense exercise. The BIA method measures impedance, i.e., resistance value, caused by the difference in electrical conductivity of each type of biological tissue, e.g., fat or muscles [14]. The four-compartment model of body composition consists of the fat mass (FM) and the lean mass (FFM), which are further divided into the total body water (TBW), proteins and minerals (Figure 1) [15].

The higher the fat mass, the higher or greater the electrical impedance [16]. BIA involves evaluating the electrical resistance of the body using the surface electrodes connected to a computer analyser, while a weak electric current (<1 mA) is applied. BIA devices may vary depending on the electrode characteristics (number, type and location), the frequency of electric current (single or multiple frequencies), and the position of the body during measurements [16]. This standardised method works by measuring the electrical resistance of the body tissues at different frequencies in relation to the body fluid volume (Figure 2) [12,17].

The electrical conductivity is proportional to the amount of water or electrolytes, which allows BIA to quantify intracellular (ICW) and extracellular water (ECW) as well as the fat and muscle mass. This method is used to assess body composition [12,14]. The devices can operate using a single frequency of 50 kHz or multi-frequency (from 5 to 1000 kHz) body composition analysis [18,19]. The applied electric current may be segmented, spectral or operate on multi-bioelectrical frequencies. The high frequency current flows through the whole body water, while the low frequency current cannot penetrate the cell membranes and therefore it flows only through the extracellular water [18]. An important issue in the clinical practice is the assessment of the relationship between the patients’ nutritional status and their hydration status. The measurement of hydration status can be based on full body measurements (wrist to ankle) or segment measurements (arm, trunk and leg) [20,21]. For many years, this method has been recognised as a safe diagnostic technique for measuring the distribution of adipose tissue and the state of hydration. Assessing body composition with the use of BIA appliances is common in health and fitness centres and research studies [16]. BIA is used in many branches of medicine (Figure 3).

It can be successfully used in both malnourished [22,23,24,25], and obese people [13,16,17,18,19,20,21,22,23,24,25,26,27,28]. However, for patients with severe abdominal obesity, percentage of body fat may be overestimated [29]. Its safety has been demonstrated in many studies of patients with renal disease, including haemodialysis and transplant patients [30,31,32,33,34,35]. BIA has also been used in patients suffering from diabetes [36], also with concomitant non-alcoholic fatty liver disease (NAFLD) [37,38]. Testing body composition with bioimpedance may partially explain why non-obese people are still at risk of developing NAFLD [39]. Literature reports in the field of neurology indicate the possibility of using bioimpedance in assessing the state of hydration in patients, especially after ischemic stroke [40,41]. BIA measurements can also be used in non-invasive assessments in case of chronic vestibular disorders [42]. This method was also widely used in patients with heart disease [43], including heart failure [44,45,46]. The physical properties of BIA, its measurement variables and their clinical significance, have been well described in many previously published reports [5,47].

Our interest in reviewing the use of the BIA method in obstetrics and perinatology stems from our own use of this method to assess the body composition and hydration status in postpartum women diagnosed with gestational diabetes mellitus (GDM) or excessive gestational weight gain (EGWG) in pregnancy [48,49,50,51].

The purpose of this paper is to provide a review of different applications of BIA in pregnant and postpartum women. Scientific articles in the Pubmed database, Wiley Online Library and Google Scholar on the use of BIA in pregnancy and puerperium have been reviewed.

## 2. Pregnancy

Even though BIA cannot distinguish between the maternal and foetal tissues [52], it is considered to be a quick, pregnancy-safe and user-friendly technique. However, in spite of the advantages of BIA, the abnormal fluid distribution during pregnancy renders different BIA methods either inappropriate or in need of further validation [8]. The whole-body impedance is mainly predicted by the impedance in the limbs [53]; however, during pregnancy a large amount of water is located in the trunk [54]. Thus, a segmental impedance measurement might be advantageous in pregnant women [55], particularly in late pregnancy.

However, there are methodological issues regarding limitations of the use of BIA in pregnancy. Total body water plays a key role in determining body composition, thus appropriate preparation is needed to minimize the effect of daytime total water change in the body. This “pretest” preparation involves performing the BIA procedure always at the same time of day, with no food or beverage intake for 4 h before testing and with the bladder emptied immediately before the measurement. All these requirements may be a serious disadvantage and constitute significant limitations for clinicians who wish to apply the BIA test in their pregnant patients [56].

A complete body change during pregnancy is a potential confounding factor affecting the results of BIA measurements. In the work of Sween et al., pregnancy-specific equations were used during the BIA measurements [57] and they were validated by Van Loan et al. [58] and Lukaski et al. [59].

BIA has been used in various clinical situations and pre-pregnancy and pregnancy conditions, including oedema [60], GDM [48,50,51,61,62], excessive gestational weight gain (EGWG) [2,3,48,51,52,55,57,62,63,64,65], preeclampsia (PE) [57,60,66,67,68,69,70,71,72,73,74], gestational hypertension (GH) [73,75] or hyperemesis gravidarum [76]. Hashimoto et al. compared the impedance analysis in healthy women and in women undergoing haemodialysis whose management of bodily fluids during pregnancy is known to be complicated [35]. Since radiographic examination for the evaluation of cardiothoracic ratio cannot be performed in pregnant women, the alternative use of BIA facilitates the measurement of body fluid volume and setting the corresponding dry weight during dialysis. According to recent studies, the live birth rate is gradually increasing as dialysis therapy advances [77,78]. That is why it is essential that the course of dialysis should be adjusted to each patient. Regardless of the course of pregnancy, the pre-dialysis levels of TBW and ICW in the pregnant woman on dialysis revealed no significant alterations. During the course of pregnancy, ECW tended to rise modestly. FFM, TBW and ECW all increased throughout late pregnancy [35].

The ICW/ECW ratio can also be used to diagnose polyhydramnios as it is supposedly low before the onset of the disorder [75,79,80].

### 2.1. Weight Gain in Pregnancy and Gestational Diabetes Mellitus

The BIA method can be successfully used to study the effect of excessive gestational weight gain in pregnancy on the development of obstetric complications. The incidence of EGWG in pregnancy is increasing worldwide and it is associated with complications of pregnancy, including GDM, PE, preterm labour, foetal macrosomia and obesity in the offspring [63]. GWG is positively associated with the fat mass gain but not fat-free mass. FFM includes the mass of TBW, bone, protein, and non-bone mineral mass. Unfortunately, FM and FFM cannot be divided into the maternal and foetal units by means of this method. In clinical trials, the main goal of the maternal body composition assessment is to evaluate changes in FM and FFM before, during and after pregnancy [63]. However, a study by Zhang et al. showed that all indicators of BIA (total body water, fat mass, fat-free mass, percent body fat, muscle mass, visceral fat levels, proteins, bone minerals, basal metabolic rate and lean trunk mass), age, weight and BMI were risk factors that significantly increased the occurrence of GDM [81]. In the case of pregnant women, an increase in TBW, FM, FFM, body cell mass (BCM) and ECW was observed. It is related to the physiological changes taking place during pregnancy, adapting the woman’s body to the developing pregnancy and subsequent lactation [66,82]. These changes are progressive and increase as pregnancy continues. From early to late pregnancy, the rate of fat accumulation was comparable [13]. An increase in the fluid retention as well as an increase in the blood volume are both reflective of elevated levels of TBW and ECW [82]. However, during pregnancy, the mean percentage of TBW decreases [13]. Overweight or obese women present significantly lower percentages of TBW. Body fat percentage (BFP), FM and TBW have been observed to be considerably greater in obese women [13]. In healthy women, FM increases during pregnancy despite a slight increase in total energy expenditure and no change is observed in the energy intake [64].

It seems that BIA has a better prognostic potential for gestational and post-partum outcomes than BMI [48,50,65]. With the metric system, the formula for BMI is weight in kilograms divided by height in meters squared. However, Asian individuals have more body fat than Caucasians with the same BMI values [83,84]. Additionally, it is known that the correlation between BMI and body fat content is age and gender dependent. BMI is a crude marker for general fat, and cannot distinguish between the fat and lean body masses [85,86]. In the case of women with a high content of the muscle tissue, BMI does not fulfil its function properly. BMI, along with gestational weight gain, inform about the real nutritional status in pregnancy, however, these parameters do not provide information regarding the distribution of fat [65]. The body fat composition, on the other hand, can be assessed in detail with the use of BIA. Recently, the fat and free-fat masses are known to be more accurate predictors of the maternal nutritional status than BMI [65]. The studies have shown that obesity and EGWG are associated with more frequent adverse maternal and neonatal complications (Table 1) [48,49,51,87].

Higher rates of newborns with a low Apgar score have been observed in obese women compared to women with normal BMI [87,88]. The studies showed that babies born to obese women had an increased risk of admission to the Intensive Care Unit for newborns and their birth weight was above 4000 g [87,89]. The BFP is considered to be a stronger predictor of GDM than BMI [87]. The mean BFP varies significantly during pregnancy [13].

A study by Zhang et al. revealed that bone minerals in early pregnancy were a significant risk factor for GDM [81]. Pathological changes in the maternal TBW detected by means of BIA measurements have been related to gestational maladaptation [90]. It is worth noting that the BIA measurements must be interpreted considering the background of adequate reference values for the population of interest as both bioelectrical properties and their relationship to the body composition are affected by the height, weight, hydration status and stage of life of individual women [67,90].

The amount and composition of a healthy GWG varies greatly among women who are underweight, normal weight, overweight and obese [1]. Studying the mother’s body composition and correlating it with her energy balance could facilitate the development of dietary recommendations for women that would help to ensure adequate weight gain during pregnancy. Obesity is a major risk factor for cardiovascular disease [91]. In the studies by Piuri et al. it has been shown that women with hypertensive disorders caused by overweight and obesity had increased TBW and ECW already in early pregnancy [82]. In contrast, women who delivered Small-for-Gestational-Age (SGA) newborns, especially when associated with foetal growth restriction, were more likely to have lower TBW and ECW values [82]. However, the studies were conducted on a small number of women.

EGWG and GDM can have an important effect on the foetal development, which is manifested by an increased predisposition to obesity, insulin resistance, diabetes, hypertension and other diseases in the later life of the offspring. The “foetal programming” hypothesis suggests that adverse effects early in the development lead to permanent changes in the structure, physiology and metabolism of the foetus [92,93,94,95]. These processes make the offspring vulnerable to obesity and related diseases such as diabetes mellitus type 2 (T2DM), hypertension, cardiovascular disease and others throughout their lives, from childhood to adulthood [92,93,96]. These adverse effects are especially noticeable in the presence of maternal nutritional imbalance and metabolic disorders during pregnancy as well as redox dysregulation in the mother–placenta–foetus unit [93].

GDM causes both short- and long-term complications for mothers and foetuses, so it is important to identify the risk factors as early as possible and implement appropriate measures to prevent their development.

Liu et al. investigated a relationship between the body composition measured by BIA at 13 weeks of gestation and GDM diagnosed at 24–28 weeks of gestation [61]. They observed that in those pregnant women whose fat mass percentage (FMP) was above 28% the risk of developing GDM was higher than in the women who presented with normal FMP. In Wang’s studies, the percentage of body fat was the strongest risk factor for GDM after adjusting the BMI before pregnancy [62]. On the other hand, the skeletal muscle mass percentage (SMMP) was inversely related to the increased risk of developing GDM [61]. The fat mass index (FMI) in early pregnancy has been shown to be a predictor of GDM. FMI may be an indicator of the effectiveness of an intervention to reduce the risk of GDM [61]. In Balani’s research, the visceral fat mass (VFM) was found to influence the development of GDM [97]. Maternal hyperglycaemia may also be a risk factor for foetal programming, as it has been shown to reduce both foetal glucose tolerance and insulin sensitivity [93].

### 2.2. Gestational Hypertension and Preeclampsia

BIA is also used as one of the additional tests in assessing the risk of developing GH and PE [68,69,75,91]. PE is a serious disease diagnosed in 2–8% of pregnancies [98,99], and it is associated with the risk of preterm labour. For this reason, it is important that the risk of PE occurrence should be identified as soon as possible and appropriate treatment and care methods should be immediately initiated. It has been shown that BIA assessment in conjunction with other tests (e.g., haemodynamics) can be used to identify early markers of an impaired cardiovascular adaptation and body composition that may lead to complications in the third trimester of pregnancy [68]. According to several study reports, women with PE gained significantly more weight during pregnancy than women with normal blood pressure [70,71]. Maternal obesity prior to pregnancy is one of the most significant risk factors for PE [100]. A relationship has been observed between the pre-pregnancy BMI value and likelihood of PE occurrence [100,101]. Nonetheless, the use of BMI in pregnancy is very limited and the results are unreliable. BIA is an alternative method of assessing overweight and obesity on the basis of the amount of body fat. In 2015, Sween et al. investigated a relationship between the content of adipose tissue in obese women in the first trimester of pregnancy and the occurrence of PE in those mothers-to-be [57]. It was shown that an increase in the adipose tissue content increased the risk of developing PE. It was also found that body fat correlated more strongly with the risk of PE than BMI. There is a viewpoint that adipose tissue is involved in the pathophysiology of PE because obesity is related, inter alia, to oxidative stress (OS). In the case of normalised and overweight women, no correlation was observed between the increase in both BMI and adipose tissue content and the increased risk of developing PE [57].

75% of PE patients had excessive GWG [14]. The Da Silva study showed that TBW and ECW were higher in the group of women with PE, while in both absolute and relative terms the percentage of ICW was lower in this group [66,69]. In the study of Staelens, the ECW/ICW ratio is higher in preeclamptic patients compared to uncomplicated pregnancies and GH, and ICW does not differ between the groups [66].

In the study of McLennan et al., attention was drawn to the early postpartum period in women who developed PE [74]. The study found that six months after pre-eclampsia, the women presented significantly higher body weight, higher percentage of fat mass, much higher BMI as well as higher insulin resistance (HOMA-IR) and reduced HDL levels in comparison to the women with normal blood pressure during pregnancy. It was observed that women who were diagnosed with PE during pregnancy led a less active lifestyle after delivery in comparison to the women who did not develop PE [74]. However, this may be related to the higher percentage of caesarean sections in this group, which is usually associated with decreased activity after delivery [74,102]. Breastfeeding had no significant effect on total or activity-related energy expenditure in both normal blood pressure and PE groups. Women with a history of PE consumed an average of 13% less kilojoules than women with normal blood pressure during pregnancy. However, the composition of macronutrients in the diets of women from both groups was similar [74].

Meta-analyses revealed a two- to three-times higher risk of chronic cardiovascular disease and T2DM in women after PE compared to women with normotension during pregnancy [103,104,105,106].

Levario-Carrillo et al. investigated body composition in four groups of patients: women with uncomplicated pregnancy, women with GH, women with mild PE and women with severe PE [73]. It was observed that maternal body composition differed significantly in the patients with hypertensive complications during pregnancy. In the patients diagnosed with elevated blood pressure, a higher pre-pregnancy BMI was observed. 66% of women diagnosed with GH, 78% with mild PE and 70% with severe PE had increased total body water (above the 90th percentile). In all the studied groups, patients presenting with co-morbid oedema showed increased TBW. It is suggested that these data may be related to a possible inadequate water volume distribution due to the alteration in capillary permeability [73].

In a study by Yeboah et al., maternal serum leptin and lipid profile were analysed, also, body fat percentage was determined during the first trimester [72]. The study subjects were purposively selected. The authors tried to determine whether in the first trimester of pregnancy, the serum concentration of leptin and the body fat percentage (%BF) are altered in those pregnant women who subsequently develop PE, and whether these changes are significant enough to enable determining which pregnant women are likely to develop PE. The use of BIA measurements made it possible to determine %BF, whereas such obesity indicators as BMI, waist circumference, waist-to-height ratio and waste-to-hip ratio cannot determine the percentage of body fat. The study of Yeboah et al. showed that significantly higher leptin levels existed in those women who subsequently developed PE in comparison to their counterparts. Moreover, it was indicated that obese women are at a greater risk of developing PE during the course of pregnancy, which corroborates earlier studies reporting a link between obesity and an increased risk of developing PE [72].

## 3. Puerperium

It seems that the BIA method can be extensively used in puerperium. In early puerperium, after giving birth and before being discharged from the hospital, conducting BIA analysis of both body composition and hydration may give the opportunity of detecting certain abnormalities. On the one hand, the risk of complications in late puerperium, and on the other hand it can also be useful to predict the development of civilization diseases in the future. Being diagnosed with GDM, EGWG, PE and GH during pregnancy increases the risk of developing pathologies such as T2DM, obesity, cardiovascular diseases with ischaemic heart disease and strokes in the future. This is due the fact that obesity is characterised by low-grade chronic inflammation with persistently elevated OS; additionally, chronic inflammation and OS contribute to the pathophysiology of many so called civilization diseases [94]. What is more, performing BIA measurements during puerperium does not raise any ethical doubts since the patient has already given birth.

In 2018 Kimber-Trojnar et al., observed that mothers with GDM in the early puerperium, when compared with the healthy controls, presented higher levels of not only fat tissue index (FTI), which is defined as the adipose tissue mass divided by the square of the body height and expressed in units of kg/m^2^, but also of TBW and ECW, where the latter consists of the interstitial water, plasma water and transcellular water [50]. Another study of this team revealed that the lowest values of the analysed BIA parameters (i.e., TBW, ECW and FTI) were observed in the healthy study subjects. The EGWG group was characterised by the highest values regarding ICW between all the studied groups as well as higher values of lean tissue index (LTI) and body cell mass index (BCMI) in comparison to the healthy mothers. The aforementioned results suggest that the EGWG women presented a higher degree of disturbances in the hydration status and body composition in the early puerperium [51].

As a result of the research work of Bzikowska-Jura et al., who used the BIA method, it became known that it is not the diet but the composition of the maternal body that may be related to the total protein concentration in breast milk. A positive correlation with the maternal weight, BMI, FM and muscles, and a negative correlation with TBW were reported [107]. Maternal BMI and obesity were positively related to the protein content of milk. Kugananthan et al. [108] and Quinn et al. [109], reported that a higher maternal FM percentage was associated with higher protein concentrations in milk. Higher maternal weight, BMI, FFM, FFM index, and FM index were associated with higher concentrations of whey protein [110].

## 4. Conclusions

Advanced body composition assessment methods in pregnant and puerperal women are an important tool in revealing any interactions between FM and inauspicious effects on the mother and her offspring. Body composition assessment helps to better understand the significance of maintaining appropriate body weight during pregnancy, which will in turn help reduce the incidence of diseases related to EGWG in women and their children. BIA could also help to identify patients at increased risk of developing different clinical phenotypes of hypertensive diseases of pregnancy in early gestation. Evaluation of the body composition can also provide important clues for diagnosis of GDM in pregnant women with a low risk of this disease. The use of the BIA method in puerperium may help in the prediction of civilization diseases in the future, such as obesity, T2DM, cardiovascular diseases with ischemic heart disease and strokes. However, more research is needed to shed more light on the correlation between the body composition and pregnancy development, which could thoroughly explain pathologies of pregnancy and puerperium.

## Figures and Tables

**Figure 1 diagnostics-11-01370-f001:**
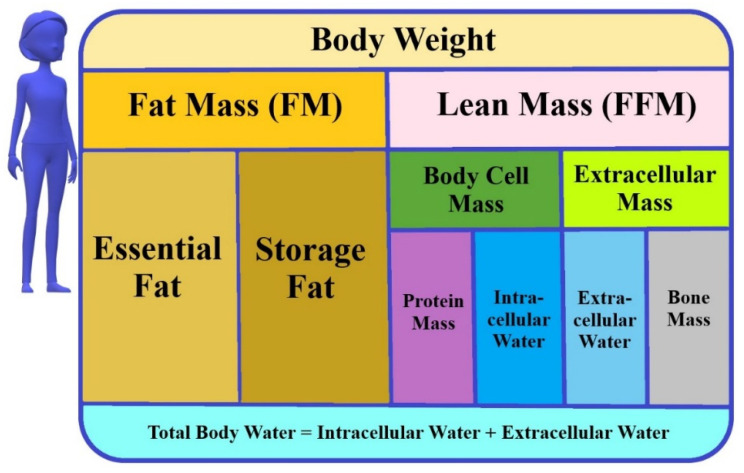
The model of body composition (used in BIA).

**Figure 2 diagnostics-11-01370-f002:**
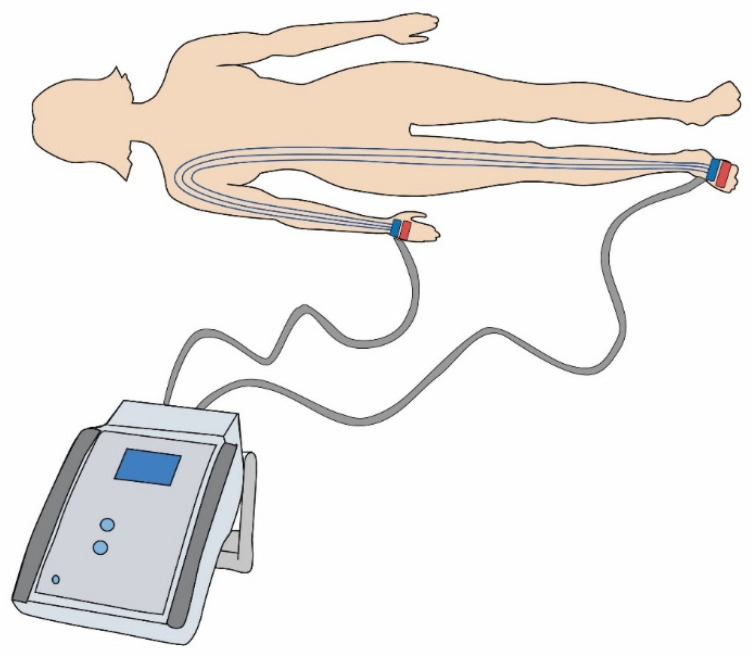
Diagram of surface electrodes connected to the bioimpedance analysis device.

**Figure 3 diagnostics-11-01370-f003:**
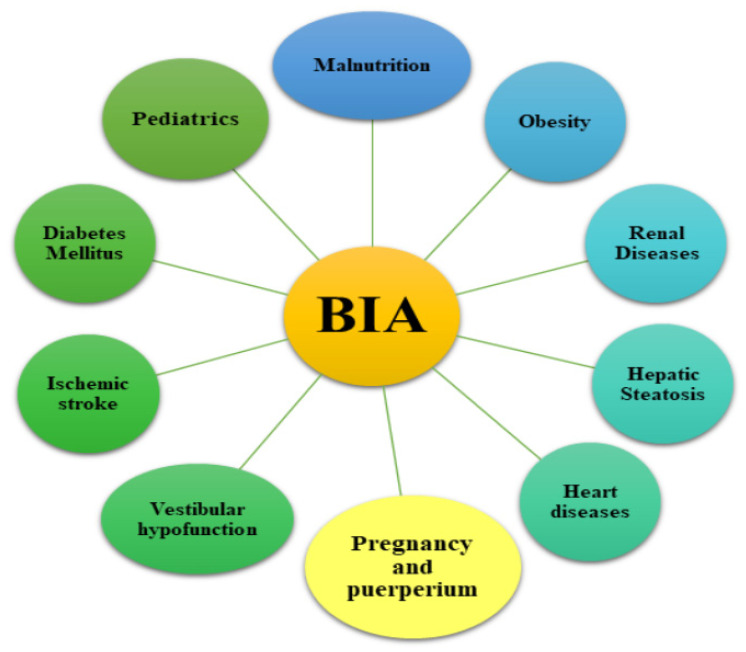
Possible application of bioelectrical impedance in healthcare. BIA—Bioimpedance analysis.

**Table 1 diagnostics-11-01370-t001:** Obesity and excessive gestational weight gain—the risk of development of the diseases for the mother and her offspring.

	Early Risk	Long-Term Risk
**Mother**	gestational diabetes mellitus	diabetes mellitus type 2
	gestational hypertension and preeclampsia	cardiovascular diseases
	caesarean delivery and vacuum/forceps delivery	
**Offspring**	prematurity	cardiovascular diseases
	macrosomia	obesity
	Intensive Neonatal Care Unit admission after delivery	hypertension
	lower Apgar score	insulin resistance and diabetes mellitus type 2

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
