# Peer review of "The Possibility of Using Bioelectrical Impedance Analysis in Pregnant and Postpartum Women"

_diagnostics, 2021, doi:10.3390/diagnostics11081370_

Round 1
Reviewer 1 Report
This is a rather new and non-mainstream method to be experimentally used to predict gestational complications as well as in the postpartum recovery period to predict the risk of developing metabolic and chronic illness in the future which may onset at middle age adults. Nowadays obesity is proved to relate to the future risk of medical conditions, most of the research group claimed it may be due to chronic inflammation, thus the authors should strengthen their manuscript by adding the possible underlying cause of relationship between obesity and future so called civilization diseases. Then they should review as they had done partially that this method is better than BMI.
The authors should utilize illustrations to clearly depict the relationship among the markers they discussed in the manuscript, such as total body water, body weight, .... to the diseases with which existing and the emerging modalities to detect these relationships.
Also the authors should use more tables to clearly guide the readers what are they points want the readers to know, otherwise the whole work looks like a very lengthy description paper, which will not attract the readers to read.
Meanwhile, if the authors had some preliminary data of their own or what they had published it is better to clearly state them in the introduction to justify why they can review this topic.
Author Response
Dear Reviewer 1,
We would like to resubmit our manuscript entitled “The possibility of using bioelectrical impedance analysis in pregnant and postpartum women”. We appreciate your valuable remarks and hope that the quality of our manuscript is going to meet your expectations now that we have made some suggested alternations.
We have rearranged our paper in accordance with your valuable comments and suggestions. The manuscript has been checked by a native-speaker.
This is a rather new and non-mainstream method to be experimentally used to predict gestational complications as well as in the postpartum recovery period to predict the risk of developing metabolic and chronic illness in the future which may onset at middle age adults.
Thank you very much for your valuable and thoughtful comments. We find all your remarks spot on therefore we have made a point-by-point correction of the manuscript according to your suggestions.
Nowadays obesity is proved to relate to the future risk of medical conditions, most of the research group claimed it may be due to chronic inflammation, thus the authors should strengthen their manuscript by adding the possible underlying cause of relationship between obesity and future so called civilization diseases.
We agree wholeheartedly with the Reviewer`s opinion regarding that obesity is proved to relate to the future risk of medical conditions. The studies have shown that obesity and overweight are associated with more frequent adverse maternal and neonatal complications. We have presented the risk of development of the diseases for the mother and her offspring due to the obesity and excessive gestational weight gain (EGWG) as follows:
“However, a study by Zhang et al. showed that all indicators of BIA (total body water, fat mass, fat-free mass, percent body fat, muscle mass, visceral fat levels, proteins, bone minerals, basal metabolic rate, lean trunk mass), age, weight and BMI were risk factors that significantly increased the occurrence of GDM [81]. [….]
The studies have shown that obesity and EGWG are associated with more frequent adverse maternal and neonatal complications (Table 1) [48,49,51,87].
Table 1. Obesity and excessive gestational weight gain - the risk of development of the diseases for the mother and her offspring.
|
|
Early risk |
Long-term risk |
||
|
Mother |
gestational diabetes mellitus |
diabetes mellitus type 2 |
||
|
gestational hypertension and preeclampsia |
cardiovascular diseases |
|||
|
caesarean delivery and vacuum/forceps delivery |
|
|||
|
Offspring |
prematurity |
cardiovascular diseases |
||
|
macrosomia |
obesity |
|||
|
Intensive Neonatal Care Unit admission after delivery |
hypertension
|
|||
|
lower Apgar score
|
insulin resistance and diabetes mellitus type 2 |
|||
Higher rates of newborns with a low Apgar score have been observed in obese women compared to women with normal BMI [87,88]. The studies showed that babies born to obese women had an increased risk of admission to the Intensive Care Unit for newborns and their birth weight was above 4,000 g [87,89]. The BFP is considered to be a stronger predictor of GDM than BMI [87]. The mean BFP varies significantly during pregnancy [13].”
References:
- Bai, M.; Susic, D.; O’Sullivan, A. J.; Henry, A. Reproducibility of Bioelectrical Impedance Analysis in Pregnancy and the Association of Body Composition with the Risk of Gestational Diabetes: A Substudy of MUMS Cohort. J. Obes. 2020, 2020.
- Zhang, R.Y.; Wang, L.; Zhou, W.; Zhong, Q.M.; Tong, C.; Zhang, T.; Han, T.L.; Wang, L.R.; Fan, X.; Zhao, Y.; Ran, R.T.; Xia, Y.Y.; Qi, H.B.; Zhang, H.; Norris, T.; Baker, P.N.; Saffery, R. Measuring Maternal Body Composition by Biomedical Impedance Can Predict Risk for Gestational Diabetes Mellitus: A
- Zhao, Y.N.; Li, Q.; Li, Y.C. Effects of Body Mass Index and Body Fat Percentage on Gestational Complications and Outcomes. J. Obstet. Gynaecol. Res. 2014, 40, 705–710.
- Briese, V.; Voigt, M.; Wisser, J.; Borchardt, U.; Straube, S. Risks of Pregnancy and Birth in Obese Primiparous Women: An Analysis of German Perinatal Statistics. Arch. Gynecol. Obstet. 2011, 283, 249–253.
- Henson, M.C.; Castracane, V.D. Leptin in Pregnancy. Biol. Reprod. 2000, 63, 1219–1228.
Additionally, we discussed the problem of fetal programming and its impact on the health of an offspring in childhood and adulthood (lines 219-228 and 242-244). We also paid attention to the correlation between low-grade chronic inflammation, obesity and the occurrence of civilization diseases in the future (lines 322-325).
Lines 219-228:
“EGWG and GDM can have an important effect on the foetal development, which is manifested by an increased predisposition to obesity, insulin resistance, diabetes, hypertension and other diseases in the later life of the offspring. The "foetal programming" hypothesis suggests that adverse effects early in the development lead to permanent changes in the structure, physiology and metabolism of the foetus [92-95]. These processes make the offspring vulnerable to obesity and related diseases such as diabetes mellitus type 2 (T2DM), hypertension, cardiovascular disease and others throughout their lives, from childhood to adulthood [92,93,96]. These adverse effects are especially noticeable in the presence of maternal nutritional imbalance and metabolic disorders during pregnancy as well as redox dysregulation in the mother-placenta-foetus unit [93].”
Lines 242-244:
“Maternal hyperglycaemia may also be a risk factor for foetal programming, as it has been shown to reduce both foetal glucose tolerance and insulin sensitivity [93].”
Lines 322-325:
“This is due the fact that obesity is characterized by low-grade chronic inflammation with persistently elevated OS; additionally, chronic inflammation and OS contribute to the pathophysiology of many so called civilization diseases [94].”
References:
- Kwon, E.J.; Kim, Y.J. What Is Fetal Programming?: A Lifetime Health Is under the Control of in Utero Health. Obstet. Gynecol. Sci. 2017, 60, 506–519.
- Kopp, W. How Western Diet and Lifestyle Drive the Pandemic of Obesity and Civilization Diseases. Diabetes Metab. Syndr. Obes. Targets Ther. 2019, 12, 2221–2236.
- Kopp, W. Development of Obesity: The Driver and the Passenger. Diabetes Metab. Syndr. Obes. Targets Ther. 2020, 13, 4631–4642.
- Marciniak, A.; Patro-MaÅ‚ysza, J.; Kimber-Trojnar, Å».; Marciniak, B.; Oleszczuk, J.; LeszczyÅ„ska-Gorzelak, B. Fetal Programming of the Metabolic Syndrome. Taiwan. J. Obstet. Gynecol. 2017, 56, 133–138.
- Paknahad, Z.; Fallah, A.; Moravejolahkami, A.R. Maternal Dietary Patterns and Their Association with Pregnancy Outcomes. Clin. Nutr. Res. 2019, 8, 64–73.
Then they should review as they had done partially that this method is better than BMI.
Following your advice, we have expanded the topic of the advantage of using BIA over BMI (lines 177-183) as follows:
“With the metric system, the formula for BMI is weight in kilograms divided by height in meters squared. However, Asian individuals have more body fat than Caucasians with the same BMI values [83,84]. Additionally, it is known that the correlation between BMI and body fat content is age and gender dependent. BMI is a crude marker for general fat, and cannot distinguish between the fat and lean body masses [85,86]. In the case of women with a high content of the muscle tissue, BMI does not fulfil its function properly.”
References:
- Deurenberg, P.; Bhaskaran, K.; Lian, P.L.K. Singaporean Chinese Adolescents Have More Subcutaneous Adipose Tissue than Dutch Caucasians of the Same Age and Body Mass Index. Asia Pac. J. Clin. Nutr. 2003, 12, 261–265.
- Gallagher, D.; Heymsfield, S.B.; Heo, M.; Jebb, S.A.; Murgatroyd, P.R.; Sakamoto, Y. Healthy Percentage Body Fat Ranges: An Approach for Developing Guidelines Based on Body Mass Index. Am. J. Clin. Nutr. 2000, 72, 694–701.
- Wells, J.C.K.; Fewtrell, M.S. Measuring Body Composition. Arch. Dis. Child. 2006, 91, 612–617.
- Vitale, S.G.; Corrado, F.; Caruso, S.; Benedetto, A.D.; Giunta, L.; Cianci, A.; D’Anna, R. Myo-Inositol Supplementation to Prevent Gestational Diabetes in Overweight Non-Obese Women: Bioelectrical Impedance Analysis, Metabolic Aspects, Obstetric and Neonatal Outcomes – a Randomized and Open-Label, Placebo-Controlled Clinical Trial. Int. J. Food Sci. Nutr. 2021, 72, 670–679.
The authors should utilize illustrations to clearly depict the relationship among the markers they discussed in the manuscript, such as total body water, body weight, .... to the diseases with which existing and the emerging modalities to detect these relationships.
We have presented the model of body composition used in BIA and added information in lines 49-51 as follows:
Lines 49-51:
“The four-compartment model of body composition consists of the fat mass (FM) and the lean mass (FFM), which are further divided into the total body water (TBW), proteins and minerals (Figure 1) [15].”
Following your advice, to depict the relationship among the markers to the diseases with which existing we added these lines:
Lines 137-150:
“Hashimoto et al. compared the impedance analysis in healthy women and in women undergoing haemodialysis whose management of bodily fluids during pregnancy is known to be complicated [35]. Since radiographic examination for the evaluation of cardiothoracic ratio cannot be performed in pregnant women, the alternative use of BIA facilitates the measurement of body fluid volume and setting the corresponding dry weight during dialysis. According to recent studies, the live birth rate is gradually increasing as dialysis therapy advances [77,78]. That is why it is essential that the course of dialysis should be adjusted to each patient. Regardless of the course of pregnancy, the pre-dialysis levels of TBW and ICW in the pregnant woman on dialysis revealed no significant alterations. During the course of pregnancy, ECW tended to rise modestly. FFM, TBW and ECW all increased throughout late pregnancy [35].
The ICW / ECW ratio can also be used to diagnose polyhydramnios as it is supposedly low before the onset of the disorder [75,79,80].”
Lines 160-164:
“However, a study by Zhang et al. showed that all indicators of BIA (total body water, fat mass, fat-free mass, percent body fat, muscle mass, visceral fat levels, proteins, bone minerals, basal metabolic rate, lean trunk mass), age, weight and BMI were risk factors that significantly increased the occurrence of GDM [81].”
Lines 170-173:
“However, during pregnancy, the mean percentage of TBW decreases [13]. Overweight or obese women present significantly lower percentages of TBW. Body fat percentage (BFP), FM and TBW have been observed to be considerably greater in obese women [13].”
Lines 201-202:
“A study by Zhang et al. revealed that bone minerals in early pregnancy were a significant risk factor for GDM [81].”
References:
- Bai, M.; Susic, D.; O’Sullivan, A. J.; Henry, A. Reproducibility of Bioelectrical Impedance Analysis in Pregnancy and the Association of Body Composition with the Risk of Gestational Diabetes: A Substudy of MUMS Cohort. J. Obes. 2020, 2020.
- Lee, S. Y.; Gallagher, D. Assessment Methods in Human Body Composition. Curr. Opin. Clin. Nutr. Metab. Care 2008, 11, 566–572.
- Hashimoto, S.; Maoka, T.; Yamamoto, R.; Kawashima, K.; Nishikawa, A.; Sakurada, K.; Koike, T.; Shigematsu, T. Comparison of Bioelectrical Impedance Analysis in Healthy Pregnant Women and Patient on Hemodialysis. Ther. Apher. Dial. 2021, 25, 160–165
- Gyselaers, W.; Vonck, S.; Staelens, A.S.; Lanssens, D.; Tomsin, K.; Oben, J.; Dreesen, P.; Bruckers, L. Gestational Hypertensive Disorders Show Unique Patterns of Circulatory Deterioration with Ongoing Pregnancy. Am. J. Physiol. Regul. Integr. Comp. Physiol. 2019, 316 , 210–221
- Piccoli, G. B.; Minelli, F.; Versino, E.; Cabiddu, G.; Attini, R.; Vigotti, F. N.; Rolfo, A.; Giuffrida, D.; Colombi, N.; Pani, A.; Todros, T. Pregnancy in Dialysis Patients in the New Millennium: A Systematic Review and Meta-Regression Analysis Correlating Dialysis Schedules and Pregnancy Outcomes. Nephrol. Dial. Transplant. 2016, 31, 1915–1934.
- Sachdeva, M.; Barta, V.; Thakkar, J.; Sakhiya, V.; Miller, I. Pregnancy Outcomes in Women on Hemodialysis: A National Survey. Clin. Kidney J. 2017, 10, 276–281.
- Valensise, H.; Andreoli, A.; Lello, S.; Magnani, F.; Romanini, C.; De Lorenzo, A. Multifrequency Bioelectrical Impedance Analysis in Women with a Normal and Hypertensive Pregnancy. Am. J. Clin. Nutr. 2000, 72, 780–783.
- Morita, H.; Takeuchi, K.; Funakoshi, T.; Mizutori, M.; Maruo, T. Potential Use of Bioelectrical Impedance Analysis in the Assessment of Edema in Pregnancy. Clin. Exp. Obstet. Gynecol. 1999, 26, 151–154.
- Zhang, R.Y.; Wang, L.; Zhou, W.; Zhong, Q.M.; Tong, C.; Zhang, T.; Han, T.L.; Wang, L.R.; Fan, X.; Zhao, Y.; Ran, R.T.; Xia, Y.Y.; Qi, H.B.; Zhang, H.; Norris, T.; Baker, P.N.; Saffery, R. Measuring Maternal Body Composition by Biomedical Impedance Can Predict Risk for Gestational Diabetes Mellitus: A Retrospective Study among 22,223 Women. J. Matern. Fetal Neonatal Med. 2020, 0, 1–8.
Meanwhile, if the authors had some preliminary data of their own or what they had published it is better to clearly state them in the introduction to justify why they can review this topic.
Thank you very much for your valuable and highly perceptive remarks. We explained in the Introduction why we can review this topic (lines 105-108) as follows:
“Our interest in reviewing the use of the BIA method in obstetrics and perinatology stems from our own use of this method to assess the body composition and hydration status in postpartum women diagnosed with gestational diabetes mellitus (GDM) or excessive gestational weight gain (EGWG) in pregnancy [48-51].”
References:
48. Trojnar, M.; Patro-Małysza, J.; Kimber-Trojnar, Ż.; Czuba, M.; Mosiewicz, J.; Leszczyńska-Gorzelak, B. Vaspin in Serum and Urine of Post-Partum Women with Excessive Gestational Weight Gain. Medicina 2019, 55.
- Kimber-Trojnar, Ż.; Patro-Małysza, J.; Trojnar, M.; Darmochwał-Kolarz, D.; Oleszczuk, J.; Leszczyńska-Gorzelak, B. Umbilical Cord SFRP5 Levels of Term Newborns in Relation to Normal and Excessive Gestational Weight Gain. Int. J. Mol. Sci. 2019, 20, 595.
- Kimber-Trojnar, Å».; Patro-MaÅ‚ysza, J.; SkórzyÅ„ska-Dziduszko, K. E.; Oleszczuk, J.; Trojnar, M.; MierzyÅ„ski, R.; LeszczyÅ„ska-Gorzelak, B. Ghrelin in Serum and Urine of Post-Partum Women with Gestational Diabetes Mellitus. Int. J. Mol. Sci. 2018, 19.
- Kimber-Trojnar, Å».; Patro-MaÅ‚ysza, J.; Trojnar, M.; SkórzyÅ„ska-Dziduszko, K. E.; Bartosiewicz, J.; Oleszczuk, J.; LeszczyÅ„ska-Gorzelak, B. Fatty Acid-Binding Protein 4—An “Inauspicious” Adipokine—In Serum and Urine of Post-Partum Women with Excessive Gestational Weight Gain and Gestational Diabetes Mellitus. J. Clin. Med. 2018, 7.
We would like to take this opportunity to thank you for all the valuable and highly perceptive remarks which have definitely made a substantial contribution to the quality of our paper.
Yours faithfully,
Arkadiusz Standyło
Chair and Department of Obstetrics and Perinatology, Medical University of Lublin, 20-090 Lublin, Poland
Tel: +48-81-7244-769
E-mail: a.standylo@gmail.com

Reviewer 2 Report
This review by Obuchowska et al. described and discussed the application of bioelectrical impedance analysis in pregnant and postpartum women.
This review was well written. However, some recently published literature should also be cited. For instance, Bai M, Susic D, O'Sullivan AJ, Henry A. Reproducibility of Bioelectrical Impedance Analysis in Pregnancy and the Association of Body Composition with the Risk of Gestational Diabetes: A Substudy of MUMS Cohort. J Obes. 2020;2020:3128767; Hashimoto S, Maoka T, Yamamoto R, et al. Comparison of bioelectrical impedance analysis in healthy pregnant women and patient on hemodialysis. Ther Apher Dial. 2021;25(2):160-165. doi:10.1111/1744-9987.13530
Author Response
Dear Reviewer 2,
We would like to resubmit our manuscript entitled “The possibility of using bioelectrical impedance analysis in pregnant and postpartum women”. We appreciate your valuable remarks and hope that the quality of our manuscript is going to meet your expectations now that we have made some suggested alternations.
We have rearranged our paper in accordance with your valuable comments and suggestions. The manuscript has been checked by a native-speaker.
This review by Obuchowska et al. described and discussed the application of bioelectrical impedance analysis in pregnant and postpartum women.
This review was well written. However, some recently published literature should also be cited. For instance, Bai M, Susic D, O'Sullivan AJ, Henry A. Reproducibility of Bioelectrical Impedance Analysis in Pregnancy and the Association of Body Composition with the Risk of Gestational Diabetes: A Substudy of MUMS Cohort. J Obes. 2020;2020:3128767; Hashimoto S, Maoka T, Yamamoto R, et al. Comparison of bioelectrical impedance analysis in healthy pregnant women and patient on hemodialysis. Ther Apher Dial. 2021;25(2):160-165. doi:10.1111/1744-9987.13530
Thank you very much for finding the time to read our manuscript. Thank you for considering our manuscript. We agree that these publications by Bai et al. and Hashimoto et al. are very important and valuable. In the current version we added them in references and we have expanded manuscript as follows:
Lines 42-45:
“In the study of Bai et al., an excellent test repeatability throughout pregnancy has been demonstrated [13]. In spite of this, different bioimpedance measuring devices should not be used interchangeably [13].”
Lines 137-148:
“Hashimoto et al. compared the impedance analysis in healthy women and in women undergoing haemodialysis whose management of bodily fluids during pregnancy is known to be complicated [35]. Since radiographic examination for the evaluation of cardiothoracic ratio cannot be performed in pregnant women, the alternative use of BIA facilitates the measurement of body fluid volume and setting the corresponding dry weight during dialysis. According to recent studies, the live birth rate is gradually increasing as dialysis therapy advances [77,78]. That is why it is essential that the course of dialysis should be adjusted to each patient. Regardless of the course of pregnancy, the pre-dialysis levels of TBW and ICW in the pregnant woman on dialysis revealed no significant alterations. During the course of pregnancy, ECW tended to rise modestly. FFM, TBW and ECW all increased throughout late pregnancy [35].”
Lines 168-173:
“From early to late pregnancy, the rate of fat accumulation was comparable [13]. […] However, during pregnancy, the mean percentage of TBW decreases [13]. Overweight or obese women present significantly lower percentages of TBW. Body fat percentage (BFP), FM and TBW have been observed to be considerably greater in obese women [13].”
Lines 199-200:
“The mean BFP varies significantly during pregnancy [13].”
References:
- Bai M, Susic D, O'Sullivan AJ, Henry A. Reproducibility of Bioelectrical Impedance Analysis in Pregnancy and the Association of Body Composition with the Risk of Gestational Diabetes: A Substudy of MUMS Cohort. J Obes. 2020, 2020.
- Hashimoto, S.; Maoka, T.; Yamamoto, R.; Kawashima, K.; Nishikawa, A.; Sakurada, K.; Koike, T.; Shigematsu, T. Comparison of Bioelectrical Impedance Analysis in Healthy Pregnant Women and Patient on Hemodialysis. Ther. Apher. Dial. 2021, 25, 160–165.
- Piccoli, G. B.; Minelli, F.; Versino, E.; Cabiddu, G.; Attini, R.; Vigotti, F. N.; Rolfo, A.; Giuffrida, D.; Colombi, N.; Pani, A.; Todros, T. Pregnancy in Dialysis Patients in the New Millennium: A Systematic Review and Meta-Regression Analysis Correlating Dialysis Schedules and Pregnancy Outcomes. Nephrol. Dial. Transplant. 2016, 31, 1915–1934.
- Sachdeva, M.; Barta, V.; Thakkar, J.; Sakhiya, V.; Miller, I. Pregnancy Outcomes in Women on Hemodialysis: A National Survey. Clin. Kidney J. 2017, 10, 276–281.
After reviewing the available literature again, we included some additional references:
- Lee, S. Y.; Gallagher, D. Assessment Methods in Human Body Composition. Curr. Opin. Clin. Nutr. Metab. Care 2008, 11, 566–572.
- Kimber-Trojnar, Ż.; Patro-Małysza, J.; Trojnar, M.; Darmochwał-Kolarz, D.; Oleszczuk, J.; Leszczyńska-Gorzelak, B. Umbilical Cord SFRP5 Levels of Term Newborns in Relation to Normal and Excessive Gestational Weight Gain. Int. J. Mol. Sci. 2019, 20, 595.
- Piccoli, G. B.; Minelli, F.; Versino, E.; Cabiddu, G.; Attini, R.; Vigotti, F. N.; Rolfo, A.; Giuffrida, D.; Colombi, N.; Pani, A.; Todros, T. Pregnancy in Dialysis Patients in the New Millennium: A Systematic Review and Meta-Regression Analysis Correlating Dialysis Schedules and Pregnancy Outcomes. Nephrol. Dial. Transplant. 2016, 31, 1915–1934.
- Sachdeva, M.; Barta, V.; Thakkar, J.; Sakhiya, V.; Miller, I. Pregnancy Outcomes in Women on Hemodialysis: A National Survey. Clin. Kidney J. 2017, 10, 276–281.
- Valensise, H.; Andreoli, A.; Lello, S.; Magnani, F.; Romanini, C.; De Lorenzo, A. Multifrequency Bioelectrical Impedance Analysis in Women with a Normal and Hypertensive Pregnancy. Am. J. Clin. Nutr. 2000, 72, 780–783.
- Morita, H.; Takeuchi, K.; Funakoshi, T.; Mizutori, M.; Maruo, T. Potential Use of Bioelectrical Impedance Analysis in the Assessment of Edema in Pregnancy. Clin. Exp. Obstet. Gynecol. 1999, 26, 151–154.
- Zhang, R.Y.; Wang, L.; Zhou, W.; Zhong, Q.M.; Tong, C.; Zhang, T.; Han, T.L.; Wang, L.R.; Fan, X.; Zhao, Y.; Ran, R.T.; Xia, Y.Y.; Qi, H.B.; Zhang, H.; Norris, T.; Baker, P.N.; Saffery, R. Measuring Maternal Body Composition by Biomedical Impedance Can Predict Risk for Gestational Diabetes Mellitus: A Retrospective Study among 22,223 Women. J. Matern. Fetal Neonatal Med. 2020, 0, 1–8.
- Deurenberg, P.; Bhaskaran, K.; Lian, P.L.K. Singaporean Chinese Adolescents Have More Subcutaneous Adipose Tissue than Dutch Caucasians of the Same Age and Body Mass Index. Asia Pac. J. Clin. Nutr. 2003, 12, 261–265.
- Gallagher, D.; Heymsfield, S.B.; Heo, M.; Jebb, S.A.; Murgatroyd, P.R.; Sakamoto, Y. Healthy Percentage Body Fat Ranges: An Approach for Developing Guidelines Based on Body Mass Index. Am. J. Clin. Nutr. 2000, 72, 694–701.
- Wells, J.C.K.; Fewtrell, M.S. Measuring Body Composition. Arch. Dis. Child. 2006, 91, 612–617.
- Vitale, S.G.; Corrado, F.; Caruso, S.; Benedetto, A.D.; Giunta, L.; Cianci, A.; D’Anna, R. Myo-Inositol Supplementation to Prevent Gestational Diabetes in Overweight Non-Obese Women: Bioelectrical Impedance Analysis, Metabolic Aspects, Obstetric and Neonatal Outcomes – a Randomized and Open-Label, Placebo-Controlled Clinical Trial. Int. J. Food Sci. Nutr. 2021, 72, 670–679.
- Zhao, Y.N.; Li, Q.; Li, Y.C. Effects of Body Mass Index and Body Fat Percentage on Gestational Complications and Outcomes. J. Obstet. Gynaecol. Res. 2014, 40, 705–710.
- Briese, V.; Voigt, M.; Wisser, J.; Borchardt, U.; Straube, S. Risks of Pregnancy and Birth in Obese Primiparous Women: An Analysis of German Perinatal Statistics. Arch. Gynecol. Obstet. 2011, 283, 249–253.
- Henson, M.C.; Castracane, V.D. Leptin in Pregnancy. Biol. Reprod. 2000, 63, 1219–1228.
- Kwon, E.J.; Kim, Y.J. What Is Fetal Programming?: A Lifetime Health Is under the Control of in Utero Health. Obstet. Gynecol. Sci. 2017, 60, 506–519.
- Kopp, W. How Western Diet and Lifestyle Drive the Pandemic of Obesity and Civilization Diseases. Diabetes Metab. Syndr. Obes. Targets Ther. 2019, 12, 2221–2236.
- Kopp, W. Development of Obesity: The Driver and the Passenger. Diabetes Metab. Syndr. Obes. Targets Ther. 2020, 13, 4631–4642.
- Marciniak, A.; Patro-MaÅ‚ysza, J.; Kimber-Trojnar, Å».; Marciniak, B.; Oleszczuk, J.; LeszczyÅ„ska-Gorzelak, B. Fetal Programming of the Metabolic Syndrome. Taiwan. J. Obstet. Gynecol. 2017, 56, 133–138.
- Paknahad, Z.; Fallah, A.; Moravejolahkami, A.R. Maternal Dietary Patterns and Their Association with Pregnancy Outcomes. Clin. Nutr. Res. 2019, 8, 64–73.
We would like to take this opportunity to thank you for all the valuable and highly perceptive remarks which have definitely made a substantial contribution to the quality of our paper.
Yours faithfully,
Arkadiusz Standyło
Chair and Department of Obstetrics and Perinatology, Medical University of Lublin, 20-090 Lublin, Poland
Tel: +48-81-7244-769
E-mail: a.standylo@gmail.com

Round 2
Reviewer 1 Report
The authors show their capability to improve the manuscript which is now deemed as suitable for publication.